# Rapid identification of *Biomphalaria* spp. and diagnosis of *Schistosoma mansoni* infestation using MALDI-TOF mass spectrometry

Diara Sy[1,2,3,4], Lionel Almeras[1,3,5], Adama Zan Diarra[2,3,4], Souleymane Doucoure[3], Yacine Mbere Sarr[3], Coralie L'Ollivier[1,4], Papa Mouhamadou Gaye[4], El Hadji Ibrahima Ndiaye[4], Bruno Senghor[3], Doudou Sow[6], Cheikh Sokhna[2,3], Stephane Ranque[1,4]*

**1** RITMES, Aix Marseille University, Service de Santé des Armées, Marseille, France, **2** EMR MINES Infectious, Neglected, and Emerging Diseases in the South, Aix-Marseille University, Research Institute for Development, Marseille, France, **3** EMR MINES: Infectious, Neglected, and Emerging Diseases in the South, Research Institute for Development, International Campus of the Institute of Research for Development, Université Cheikh Anta Diop de Hann, Dakar, Senegal, **4** Parasitology-Mycology, IHU Méditerranée Infection, Marseille, France, **5** Parasitology and Entomology Unit, Department of Microbiology and Infectious Diseases, Armed Forces Biomedical Research Institute, Marseille, France, **6** Department of Parasitology-Mycology, Health Sciences Training and Research Unit, Université Gaston Berger, Saint Louis, Senegal

* stephane.ranque@univ-amu.fr

## Abstract

This study explores the use of Matrix-Assisted Laser Desorption/Ionization Time-of-Flight mass spectrometry (MALDI-TOF MS) to identify and differentiate *Biomphalaria* snails infected with the parasite *S. mansoni*, which causes schistosomiasis. The study was conducted on two snail species, *Biomphalaria pfeifferi* (collected in the field in Senegal) and *Biomphalaria glabrata* (a laboratory strain). The snails were infected in the laboratory with *S. mansoni* miracidia, and their infection was confirmed by cercariae emission tests and quantitative PCR (qPCR). MALDI-TOF MS was then used to analyse proteins from infected and uninfected snails to identify spectral differences. Based on protein profiles, the results of MALDI-TOF mass spectrometry made it possible to accurately differentiate between S. mansoni-infected snails and uninfected snails. An increase in the number of peaks detected and their intensity was observed for the spectra of *S. mansoni*-infected snails compared to uninfected snails. The application of principal component analysis (PCA) to these mass spectrometry profiles confirmed the discrimination between the two groups according to their infection status. In addition, specific discriminating peaks were identified for each snail species, allowing for the distinction of infected from uninfected snails. The present study revealed, for the first time, that MALDI-TOF MS appears to be a rapid, reliable, and specific tool for the diagnosis of schistosomiasis in snails, offering promising prospects for the surveillance and control of this disease in endemic areas. However, further work is needed to establish a MALDI-TOF MS reference spectra database

Data availability statement: All relevant data generated during this study are included in the manuscript.

Funding: This work was supported by a grant from the French Government managed by the French National Research Agency (ANR) under the "Investissements d'avenir (Investments for the Future)" programme with the reference ANR-10-IAHU-03 (Méditerranée Infection), and by the Conseil Régional Provence Alpes Côte d'Azur, Contrat Plan Etat-Région, and the European funding FEDER IHUPERF. The Institut de Recherche pour le Développement (IRD) funded the EMR MINES: Infectious, Neglected, and Emerging Diseases in the South. DS received a doctoral scholarship from the Méditerranée Infection Foundation. The funders had no role in study design, data collection and analysis, decision to publish, or preparation of the manuscript.

Competing interests: The authors have declared that no competing interests exist.

specific to *Schistosoma* parasites and to standardise sample collection, storage, and preparation in order to apply this technique in the field.

## Author summary

This study investigates the use of Matrix-Assisted Laser Desorption/Ionisation Time-of-Flight mass spectrometry (MALDI-TOF MS) to identify *Biomphalaria* snails infected with the *Schistosoma mansoni* parasite, which causes schistosomiasis. Researchers focused on two snail species: *Biomphalaria pfeifferi* from Senegal and a laboratory strain of *Biomphalaria glabrata*. Infected snails were confirmed through tests that measured the emission of cercariae and quantitative PCR (qPCR). The MALDI-TOF MS analysis revealed distinct protein profiles between infected and uninfected snails. In particular, infected snails showed a higher number of spectral peaks and greater peak intensity. By applying principal component analysis (PCA) to the mass spectrometry data, the study successfully differentiated between the two groups based on their infection status. Notably, specific peaks were identified that distinguished each snail species. This research demonstrates that MALDI-TOF MS can serve as a quick and reliable method for diagnosing schistosomiasis in snails, which could enhance disease surveillance and control in affected regions. However, further work is necessary to develop a comprehensive reference database and standardise protocols for field application.

## Introduction

Schistosomiasis is a parasitic infection caused by contact with surface water contaminated by the parasite of the genus *Schistosoma* [1,2]. This parasitic disease is very serious and affects more than 190 million people worldwide [3]. Seventy-eight countries are currently facing problems resulting from human schistosomiasis, the majority of them in sub-Saharan Africa [4]. Freshwater snails can be intermediate hosts for many parasites of medical and veterinary interest [5], and the genus *Biomphalaria* contributes to the transmission of *Schistosoma mansoni*, responsible for intestinal schistosomiasis [6]. *Biomphalaria pfeifferi* is a freshwater snail which is widely distributed in sub-Saharan Africa and Madagascar, as well as in some areas of the Sahara and Southwest Asia [7]. Another species, *Biomphalaria glabrata* is native to South America [8]. Meanwhile, in Senegal, the only species of the genus *Biomphalaria* is *Biomphalaria (Bi.) pfeifferi,* an intermediate host of the parasite *S. mansoni* [9]. It is, therefore, clear that several strategies are needed to control schistosomiasis, including controlling the infestation dynamics of molluscs in endemic contact areas [10].

Several diagnostic methods for parasite surveillance have been developed to monitor the evolutionary dynamics of schistosomiasis in endemic areas [11]. These include manual cercariometry [12,13], positive phototropism [14], the superposition technique [15], and centrifugation [16]. However, to detect infestation with these methods, the

parasite must be in its final stage of evolution. Techniques such as qPCR [17] and molecular detection and identification [18] have emerged to detect parasites at an early stage. These two methods, however, cannot be implemented in the field and require significant preparation time. Recently, an innovative approach has emerged for the identification and diagnosis of pathogens and their vectors, based on protein profiling: matrix-assisted laser desorption/ionization time-of-flight mass spectrometry (MALDI-TOF MS) [19,20]. MALDI-TOF MS has proven to be a rapid, reliable, and specific method for identifying parasites, bacteria, viruses, and arthropod vectors, representing significant potential in the field of clinical microbiology and disease control [21]. However, it has also enabled us to identify the intermediate host snails of the parasite *Schistosoma* [22]. In addition, Huguenin *et al.* were able to identify and differentiate different species of parasites of the genus *Schistosoma* [23]. The primary objective of this study was to discriminate between the two species of *Biomphalaria*, *Bi. pfeifferi* and *Bi. glabrata*, using MALDI-TOF MS and then to assess whether MALDI-TOF MS could detect the *S. mansoni* parasite in the snails.

## Materials and methods

### Ethics statement

The experimental protocol was approved by the Senegal Ethics Committee (Comité National d'Ethique pour la Recherche en Santé, CNERS, Reference No. 000073/MSAS/DPRS/CNERS) and received clearance from the head of Nagoya in Senegal (Reference No. 001339). The snail infection protocol complies with national and international guidelines on animal welfare.

### Snail collection

Specimens of *B. pfeifferi* and *B. glabrata* infested with *S. mansoni* were obtained during a previous a study evaluating the compatibility of these two intermediate hosts with a strain of *S. mansoni* from Senegal [24]. The *B. pfeifferi* strain originates from northern Senegal, in the Senegal River valley area, while the *B. glabrata* strain, native to Recife, Brazil, was obtained from the Host–Pathogen–Environment Interactions (IHPE) laboratory in Perpignan, France.

### PCR-based *S. mansoni* detection

The presence of the *S. mansoni* parasite in snails was confirmed by the qPCR system as previously described [25]. Briefly, we used the forward primers SRA1 and antisense primers SRS2 and the probe SRP. The PCR mixing conditions were 20 µl of reaction mix containing 5 µl of template DNA, 3.5 µl of sterile ultrapure water, 0.5 µl of each primer, 0.5 µl of probe, and 10 µl of Master Mix (product ref). Amplification was performed on a CFX96 thermal cycler (BIO-RAD, Hercules, CA, USA). The PCR programme was initiated by a three-minute purification at 95 °C, followed by ten minutes' denaturation at 95 °C, and 40 cycles of 30 seconds of denaturation and hybridisation at 55 °C for 30 seconds, maintained at 4 °C [25]. In each run, a negative control (sterile water) and a positive control (DNA extracted from *S. mansoni* worms) were used. PCR results were considered positive if the cycle threshold (CT) was less than 35 cycles.

### MALDI TOF MS-based *S. mansoni* detection

**Snail preparation.** The soft part from the shell was carefully removed and the foot was dissected under a Leica ES2 10x/30x stereomicroscope using a sterile slide. The remaining parts of the snail were stored at -20 °C for further study. The dissected part was first rinsed with 70% ethanol and then distilled water for two minutes. It was dried, deposited on a sterile filter paper, and cut into small pieces [22]. The foot samples were then added to an extraction solution composed of a mixture of 70% formic acid (Sigma-Aldrich, Lyon, France), 50% acetonitrile (Fluka, Buchs, Switzerland), and 80% HPLC water. For each sample, 30 µL of the mixed solution and glass beads (Sigma-Aldrich, St. Louis, Missouri, USA) was homogenised using a Tissue Lyser II (Qiagen, Germany) with optimised parameters (three one-minute cycles at a frequency of 30 Hertz), as described previously [26]. All homogenates were centrifuged at 2000 g for 30 seconds, and

1.5 µL of each supernatant was spotted onto a steel target plate (Bruker Daltonics GmBh, Bremen, Germany) in four spots. One microlitre of a CHCA matrix suspension composed of saturated alpha-cyano-4-hydroxycinnamic acid (Sigma, Lyon, France), 50% acetonitrile, 2.5% trifluoroacetic acid (Aldrich, Dorset, UK), and 47.5% of HPLC grade water was directly deposited onto each spot on the target plate to allow co-crystallisation, then dried at room temperature before being inserted into the MALDI-TOF MS instrument (Bruker Daltonics) [27].

**Analysis of MALDI-TOF MS spectra.** Mass spectrometry analysis was performed using a Microflex LT MALDI-TOF mass spectrometer (Bruker Daltonics) with the Flex Control software (Bruker Daltonics). Measurements were performed in linear positive ion mode in a 2 kDA–20 kDa mass range. Each spectrum corresponds to ions obtained from 240 laser shots performed in six regions of a single spot. Spectral profiles were visualised using Flex Analysis, version 3.3, and then exported to the MALDI Biotyper (Bruker Daltonics) software version 3.0 and ClinProTools v.2.2 (Bruker Daltonics) for further data processing (smoothing, baseline subtraction, peak selection). The reproducibility of MS spectra was assessed by comparing the Main Spectrum Profile (MSP) obtained from the four spots of each sample with the MALDI Biotyper (Bruker Daltonics). The composite correlation index (CCI) tool of the MALDI Biotyper was also used to assess spectral variations within and between snails which were infected with *S. mansoni* and those that were not [28]. In addition, ClinProTools was used to identify discriminatory peaks between infected and non-infected snails not only within species but also between the two species. The most discriminatory common peaks between infected and non-infected snails were analysed with ClinProTools to compare the two groups and estimate the impact of infestation. The ClinProTools default parameters for spectrum preparation were applied as described by 29 [29]. Based on the peak list obtained per species, the five and eight most intense peaks of snails infected by *Bi. pfeifferi* and *Bi. glabrata*, respectively were selected for inclusion in the genetic algorithm (GA) model. The peaks selected by the operator provided a recognition capacity (RC) value related to the highest cross-validation (CV) value. The presence or absence of all discriminatory peak masses generated by the GA model was checked by comparing the average spectra of each species per body part.

**DNA sequence-based snail species confirmation.** To confirm the identification of *Biomphalaria* species, we sequenced three infected and three uninfected specimens of each species that displayed a good MALDI-TOF MS spectrum and a high MALDI Biotyper (Bruker Daltonics) log score value (LSV) ≥ 2. To identify the snail species, we sequenced the following gene fragments: COX (CO1 (LC1490): GGTCAACAAATCATAAAGATATTGG; CO2 (HCO2198): TAAACTTCAGGGTGACCAAAAAATCA); ITS (ETTS1: TGCTTAAGTTCAGCGGGT; ETTS10: GCATACTGCTTTGAACATCG) [30], and 18S (18S F: AGTATGGTTGCAAAGCTGAAACTTA; 18S R: TACAAAGGGCAGGGACGTAAT) [31], as previously described. To confirm the amplification of DNA, the PCR products were run on a 1% agarose gel at 180 V, 400 mA for 15 minutes. For a PCR to be considered positive, the detection of a 363-base pair band was required. Positive samples were purified and sequenced using the same primers with BigDye version 1-1 Cycle Sequencing Ready Reaction Mix (Applied Biosystems, Foster City, CA) and run on an ABI 3100 automated sequencer (Applied Biosystems). Sequences were assembled and corrected using the ChromasPro (version 1.34) software (Technelysium Pty. Ltd., Tewantin, Australia) and identified using BLAST queries (https://blast.ncbi.nlm.nih.gov/Blast.cgi) against the NCBI GenBank nucleotide database.

## Results

### Snail species DNA sequence-based confirmation

Seven snails specimens, five of which were infected with *S. mansoni,* and three uninfected snails of each species, were subjected to standard PCR and sequencing using three different genes (Cox, ITS and 18S). BLAST analysis of *B. pfeifferi* and *B. glabrata* sequences showed coverage percentages of 98% to 100% and identity percentages ranging from 97% to 100% with the sequences of their counterparts in GenBank (Table 1).

**Table 1. Molecular identification of *Biomphalaria* snails.**

| | COX | | ITS | | 18S | |
|---|---|---|---|---|---|---|
| *Biomphalaria* identification | % identity | Accession | % identity | Accession | % identity | Accession |
| *Bi. pfeifferi* | 97 - 99.96 | DQ084831.1 | 98.71 - 99.81 | MG461588.1 | 99.61 - 100 | MG461588.1 |
| *Bi. grabrata* | 98.32 - 99.84 | OQ954055.1 | 97.22 - 99.63 | OX365789.1 | 99.81 - 100 | OX365789.1 |
| *Bi. pfeifferi* | 97 - 99.96 | DQ084831.1 | 98.71 - 99.81 | MG461588.1 | 99.61 - 100 | MG461588.1 |
| *Bi. grabrata* | 98.32 - 99.84 | OQ954055.1 | 97.22 - 99.63 | OX365789.1 | 99.81 - 100 | OX365789.1 |

## MALDI-TOF MS spectra analysis

The MALDI-TOF MS spectra of uninfected *Biomphalaria* snails were selected to assess the ability of this tool to discriminate between the two species. We obtained high intensity spectra, i.e., greater than 2000 au, when 20 uninfected *Biomphalaria* snails were subjected to MALDI TOF MS. These spectra appeared visually reproducible for each species (Fig 1A). To confirm the reproducibility of the spectra, cluster analyses were performed, resulting in the MSP dendrogram (Fig 1B). Three specimens of each species were selected. The clustering of specimens of the same species on the same branch with a high distance between them and the absence of intertwining species demonstrate the reproducibility and specificity of each *Biomphalaria* species protein profile.

(1)

(2)

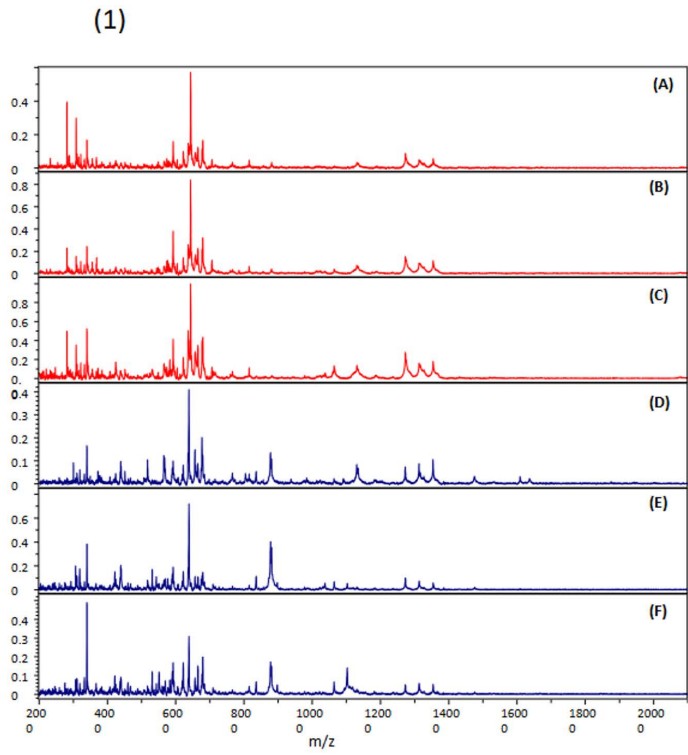
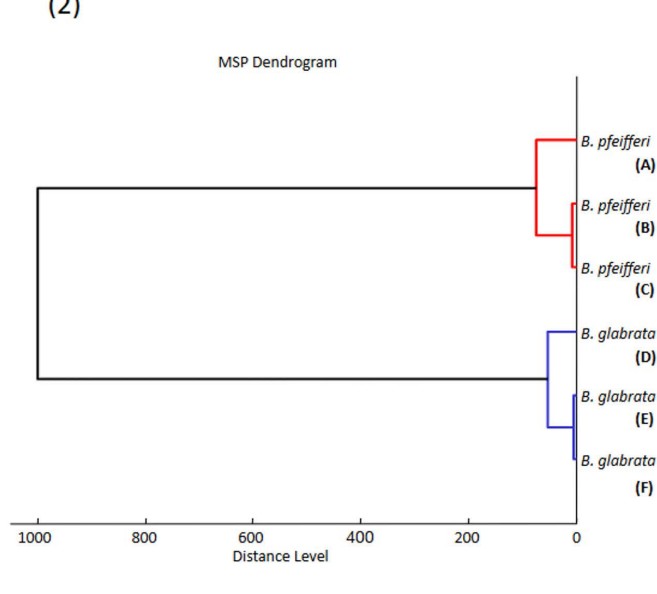

**Fig 1. Identification of *Biomphalaria* by MALDI-TOF MS profiling. (1)** Comparison of MALDI-TOF MS spectra obtained from the snails feet of three distinct specimens of *Bi. pfeifferi* (A, B, C) and *Bi. glabrata* (D, E, **F)**. (m/z, mass-to-charge ratio). **(2)** MSP dendrogram of the MALDI-TOF MS spectra of three specimens per species. The dendrogram was created using the Biotyper v3.0 software, and the distance units correspond to the relative similarity of the MS spectra.

Since the correct classification of the snail species relies primarily on the intensity of the resulting MS spectra, the five most intense peaks were selected from the uninfected *Biomphalaria* snails. This resulted in a total of eight peaks discriminating between the *Biomphalaria* species (Table 2). These eight peaks were subjected to the genetic algorithm (GA) model of ClinProTools 2.2 software. The combination of the presence or absence of these five most intense peaks displayed Cross-Validation (CV) and Recognition Capacity (RC) values equal to 100%. These five peaks were also evaluated against the infested groups. This enabled us to determine the percentage of compatibility between the groups. Values of 84% and 80% were obtained for infested *Bi. glabrata* and infested *Bi. pfeifferi,* respectively.

The two *Biomphalaria* species formed two distinct MALDI-TO spectra groups. In Fig 2, the infected group is in green and the uninfected group is in red. The separation of infected and uninfected snails was much clearer in the *Bi. glabrata* species, with ~85% of variance being explained, while only ~80% of the variance was explained in the *Bi. pfeifferi* species.

Within the *Bi. glabrata* snail group, 16 peaks differentiated snails infected with the *S. mansoni* parasite from uninfected snails. Statistical analyses showed a *P*-value less than.005 ($P < .005$) in the PTTA Analysis of Variance test (Table 3). This indicates that these peaks are informative and discriminatory on the infestation parameter. Using ClinProTools software, principal component analysis generated a cross-validation percentage of 97.67% and a recognition ability of 100%.

Within the *Bi. pfeifferi* snail group, 17 peaks distinguished snails infected with the *S. mansoni* parasite from uninfected snails. Statistical analyses showed highly significant *P* values (Table 4) in the PTTA Analysis of Variance test. This indicates that these peaks are informative and discriminatory on the infestation parameter. Using ClinProTools software, principal component analysis generated a cross-validation percentage of 95.32% and a recognition ability of 98.32%.

## Discussion

First, MALDI TOF MS made it possible to discriminate between two snail species of the same genus. Specific and reproducible spectra were obtained for each species with 100% reliability. In addition, we identified *Bi. glabrata* for the first time, using MALDI TOF MS with good quality spectra (LSV > 2). This confirms the efficiency and reliability of MALDI-TOF in identifying snail species, as confirmed by Hamlili *et al*. [22] in their previous study.

Second, we were able to clearly and accurately discriminate between infected and uninfected *Biomphalaria* snails using MALDI-TOF MS. MALDI-TOF, initially limited to identifying intermediate host snails [22], is capable of differentiating parasite infestations within the host. Since this device is a tool based on the ionisation of an organism's proteins [28], MALDI-TOF detected the presence of other proteins on samples that were positive for cercariae emission and real-time PCR, which is different from the proteins detected in uninfected snails. This resulted in a difference in spectra between the

**Table 2. List of the top eight mass peaks per *Biomphalaria* species using feet as biological material.**

| Average peak intensity (arbitrary unit)* | | | |
|---|---|---|---|
| MS peak number (§) | m/z (Da) | *Bi. glabrata* | *Bi. pfeifferi* |
| 1 | 2814.36 | 2.72 | **18.58** |
| 2 | 3400.21 | **22.46** | **14.63** |
| 3 | 5925.07 | **10.1** | **16.12** |
| 4 | 6221.43 | **10.33** | 6.96 |
| 5 | 6366.49 | 8.46 | **13.39** |
| 6 | 6387.48 | **19.09** | 11.67 |
| 7 | 6438.18 | 3.35 | **33.66** |
| 8 | 6794.58 | **10.51** | 12.51 |

§ List of MS peaks used to distinguish *Biomphalaria* species, based on analysis by the ClinProTools genetic algorithm model.

* The top five mass peaks per *Biomphalaria* species that discriminate between the two species are shown in bold. Da: Daltons; m/z: mass/charge.

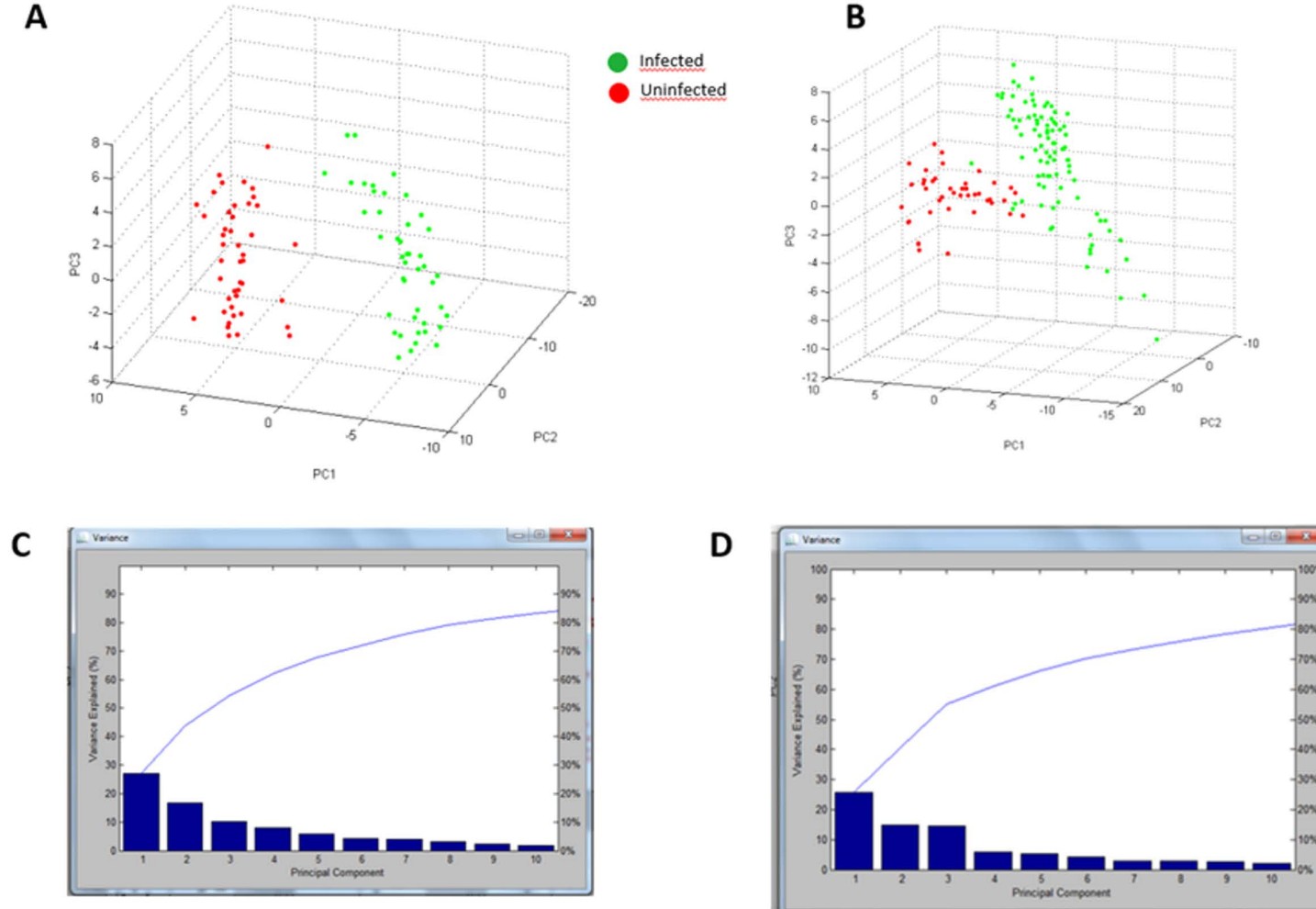

**Fig 2. Comparison of the different spectra obtained from the two *Biomphalaria* species, infected or not infected by *S. mansoni*, using principal component analysis (PCA). (A)**. *Bi. glabrata* infected (n = 20, green dots) or not infected (n = 10, red dots) with *S. mansoni*. **(B)**. *Bi. pfeifferi* infected (n = 20, green dots) or not infected (n = 10, red dots) with *S. mansoni*. **(C)**. Expression curve of variance versus principal component of *Bi. glabrata*. **(D)**. Expression curve of variance versus principal component of *Bi. pfeifferi*..

group of infected and uninfected snails. Thus, the visualised spectra confirm a difference in protein expression between infected *Biomphalaria* and uninfected *Biomphalaria*. The reproducibility in the spectra of infected snails indicates the presence of the same proteins in these snails. The same observations were made in the uninfected snails. Given that there is no MALDI-TOF spectra database to identify *Schistosoma* parasites, the parasite species remain to be identified.

The validation percentages and intra-species recognition capacity obtained confirm that MALDI-TOF is indeed capable of differentiating *Schistosoma* infestation within the same species. On the one hand, we observed different peaks, which indicate the expression of distinct proteins, and on the other we observed similar peaks in both infected and uninfected groups. In this case, the distinction lies in the intensity and regularity of expression in all individuals. In other words, the presence of the parasite can modulate the expression of certain proteins in the snail, thus making it possible to differentiate between infected snails and those that are not. Similarly, it has been demonstrated that MALDI-TOF MS could differentiate filaria-infected mosquitoes from uninfected mosquitoes, with a specificity of 100% and a sensitivity of 92% [32].

**Table 3. The 16 most discriminating MALDI-TOF MS peaks, selected with ClinProTools software, discriminating infected _Bi. glabrata_ snails from uninfected ones.**

| MS peak number§ | m/z (Da) | PTTA | *Bi. glabrata* neg (BgN) | *Bi. glabrata* pos (BgP) | Ratio BgP/BgN |
|---|---|---|---|---|---|
| 1 | 2072.63 | <0.000001 | 2.12 | 5.36 | 2.53 |
| 2 | 2217.21 | <0.000001 | 2.42 | 5.41 | 2.24 |
| 3 | 2304.48 | <0.000001 | 2.22 | 5.33 | 2.40 |
| 4 | 2399.19 | <0.000001 | 2.49 | 9.82 | 3.94 |
| 5 | 3628.45 | <0.000001 | 1.41 | 15.79 | 11.20 |
| 6 | 3644.45 | <0.000001 | 1.63 | 6.33 | 3.88 |
| 7 | 4802.11 | <0.000001 | 1.02 | 35.53 | 34.83 |
| 8 | 4818.03 | <0.000001 | 1.11 | 10.15 | 9.14 |
| 9 | 5175.48 | <0.000001 | 5.13 | 2.29 | 0.45 |
| 10 | 5311.79 | <0.000001 | 3.42 | 1.28 | 0.37 |
| 11 | 6387.3 | <0.000001 | 19.09 | 6.13 | 0.32 |
| 12 | 8355.94 | <0.000001 | 5.17 | 2.13 | 0.41 |
| 13 | 8779.91 | <0.000001 | 8.2 | 3.35 | 0.41 |
| 14 | 8804.39 | <0.000001 | 7.11 | 2.81 | 0.40 |
| 15 | 10640.59 | <0.000001 | 2.54 | 0.86 | 0.34 |
| 16 | 13129.7 | <0.000001 | 5.72 | 2.37 | 0.41 |

§ List of MS peaks used to distinguish *Biomphalaria glabrata* species, based on analysis by the ClinProTools genetic algorithm model. * The 17 main mass peaks per species of *Biomphalaria glabrata*, which allow to differentiate between the two groups of infected/non-infected snails, Da: Daltons; m/z: mass/charge; a.u.: arbitrary unit, PTTA Analysis of Variance test *P*-value

**Table 4. The 17 most discriminating MALDI-TOF MS peaks, selected with ClinProTools software, between infected _Bi. pfeifferi_ snails and uninfected ones.**

| MS peak number§ | m/z (Da) | PTTA | *Bi. pfeifferi* neg (BPn) | *Bi. pfeifferi* pos (BPp) | Ratio BPp/BPn |
|---|---|---|---|---|---|
| 1 | 4568.01 | 0.00231 | 1.69 | 5.97 | 0.28 |
| 2 | 4802.27 | 0.00834 | 1.12 | 5.28 | 0.21 |
| 3 | 4872.64 | 0.00000136 | 1.26 | 3.95 | 0.32 |
| 4 | 4889.05 | 0.0000166 | 1.98 | 4.26 | 0.46 |
| 5 | 4927.1 | 0.00000557 | 1.24 | 5.03 | 0.25 |
| 6 | 5487.01 | < 0.000001 | 3.79 | 1.61 | 2.35 |
| 7 | 5765.23 | < 0.000001 | 4.49 | 2.02 | 2.22 |
| 8 | 5829.13 | < 0.000001 | 4.82 | 1.38 | 3.49 |
| 9 | 5901.72 | < 0.000001 | 4.76 | 2.37 | 2.01 |
| 10 | 5924.64 | < 0.000001 | 16.12 | 6.72 | 2.40 |
| 11 | 6127.44 | < 0.000001 | 2.27 | 4.65 | 0.49 |
| 12 | 6437.76 | < 0.000001 | 33.66 | 8.28 | 4.07 |
| 13 | 6457.5 | < 0.000001 | 9.13 | 4.46 | 2.05 |
| 14 | 6583.76 | < 0.000001 | 7.26 | 2.99 | 2.43 |
| 15 | 7060.81 | < 0.000001 | 5.26 | 1.93 | 2.73 |
| 16 | 10641.45 | 0.00000529 | 1.83 | 0.87 | 2.10 |
| 17 | 13130.64 | < 0.000001 | 4.2 | 1.67 | 2.51 |

§ List of MS peaks used to distinguish between *Biomphalaria glabrata* species, based on analysis by the ClinProTools genetic algorithm model. * The 17 main mass peaks per species of *Biomphalaria glabrata*, which allow to distinguish between the two infected/non-infected groups, Da: Daltons; m/z: mass/charge; a.u.: arbitrary unit, PTTA Analysis of Variance test *P*-value

The slight difference in the validation percentages and the recognition capacity between *Bi. pfeifferi* and *Bi. glabrata* may be explained by the origin of the collected samples. The *Bi. pfeifferi* samples were collected in the field, while our *Bi. glabrata* strain had been raised in the laboratory for years. Thus, uncontrolled environmental conditions such as the physicochemical parameters of the water and the instability of the climate, but also water pollution can influence MALDI-TOF MS spectra and contribute to the differences between the spectra of field and laboratory snails. Indeed, MALDI-TOF MS has been found to be capable of tracing the geospatial origin of snails [33].

## Conclusion

This study demonstrates the effectiveness of Matrix-Assisted Laser Desorption/Ionization Time-of-Flight (MALDI-TOF) mass spectrometry as an innovative tool for the identification and differentiation between snails of the genus *Biomphalaria* infected with *Schistosoma mansoni*, the causative agent of schistosomiasis. By combining speed, reliability, and specificity, this method offers promising prospects for diagnosing and monitoring this parasitic disease in endemic areas, particularly in sub-Saharan Africa, where schistosomiasis is a major public health problem. The good quality MALDI-TOF spectra generated in this study will be used to enrich our in-house MALDI-TOF spectral database and further blind tests are planned to assess the capacity of MALDI-TOF MS to diagnose *Schistosoma* parasite infestations of snails collected in the field and to propose a standardised sample collection, storage, and preparation protocol. Finally, this technological advance will contribute towards developing an integrated approach aimed at reducing the public health impact of schistosomiasis and contributing to global efforts to eradicate this disease by 2030, in line with WHO targets.

## Author contributions

**Conceptualization:** Stephane Ranque.

**Formal analysis:** Diara Sy, Lionel Almeras, Adama Zan Diarra.

**Funding acquisition:** Doudou Sow, Cheikh Sokhna.

**Investigation:** Diara Sy, Yacine Mbere Sarr, Papa Mouhamadou Gaye, El Hadji Ibrahima Ndiaye.

**Project administration:** Souleymane Doucoure, Cheikh Sokhna, Stephane Ranque.

**Resources:** Souleymane Doucoure, Coralie L'Ollivier, Papa Mouhamadou Gaye, Cheikh Sokhna, Stephane Ranque.

**Software:** Bruno Senghor.

**Supervision:** Lionel Almeras, Adama Zan Diarra, Souleymane Doucoure, Coralie L'Ollivier, Doudou Sow, Cheikh Sokhna, Stephane Ranque.

**Validation:** Lionel Almeras, Adama Zan Diarra, Souleymane Doucoure, Yacine Mbere Sarr, Coralie L'Ollivier, Papa Mouhamadou Gaye, Bruno Senghor, Stephane Ranque.

**Visualization:** Diara Sy, Papa Mouhamadou Gaye.

**Writing – original draft:** Diara Sy.

**Writing – review & editing:** Lionel Almeras, Adama Zan Diarra, Souleymane Doucoure, Yacine Mbere Sarr, Coralie L'Ollivier, Papa Mouhamadou Gaye, El Hadji Ibrahima Ndiaye, Bruno Senghor, Doudou Sow, Cheikh Sokhna, Stephane Ranque.

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
