## [Decision Letter · Decision Letter 0]

11 Sep 2025

Response to Reviewers '. This file does not need to include responses to any formatting updates and technical items listed in the 'Journal Requirements' section below. * A marked-up copy of your manuscript that highlights changes made to the original version. You should upload this as a separate file labeled 'Revised Manuscript with Track Changes '. * An unmarked version of your revised paper without tracked changes. You should upload this as a separate file labeled 'Manuscript '. If you would like to make changes to your financial disclosure, competing interests statement, or data availability statement, please make these updates within the submission form at the time of resubmission. Guidelines for resubmitting your figure files are available below the reviewer comments at the end of this letter. We look forward to receiving your revised manuscript. Kind regards, Anand Setty Balakrishnan, PhDAcademic EditorPLOS Neglected Tropical Diseases Paul MirejiSection EditorPLOS Neglected Tropical Diseases

Shaden Kamhawi

co-Editor-in-Chief

Paul Brindley

co-Editor-in-Chief

**Additional Editor Comments:**Dear Authors,

**Journal Requirements:**

At this stage, the following Authors/Authors require contributions: Diara Sy, Lionel Almeras, Adama Zan Diarra, Souleymane Doucoure, Yacine Mbere Sarr, Coralie L'Ollivier, Papa Mouhamadou Gaye, Pape Ibrahima Ndiaye, Bruno Senghor, Doudou Sow, Cheikh Sokhna, and Stephane Ranque. Please ensure that the full contributions of each author are acknowledged in the "Add/Edit/Remove Authors" section of our submission form.

Potential Copyright Issues:

- Figure 1. Please confirm whether you drew the images / clip-art within the figure panels by hand. If you did not draw the images, please provide (a) a link to the source of the images or icons and their license / terms of use; or (b) written permission from the copyright holder to publish the images or icons under our CC BY 4.0 license. Alternatively, you may replace the images with open source alternatives. See these open source resources you may use to replace images / clip-art:

5) Please ensure that the funders and grant numbers match between the Financial Disclosure field and the Funding Information tab in your submission form. Note that the funders must be provided in the same order in both places as well.

**Reviewers' comments:** Reviewer's Responses to Questions

**Key Review Criteria Required for Acceptance?**

**Methods:**

-Are the objectives of the study clearly articulated with a clear testable hypothesis stated?

-Is the study design appropriate to address the stated objectives?

-Is the population clearly described and appropriate for the hypothesis being tested?

-Is the sample size sufficient to ensure adequate power to address the hypothesis being tested?

-Were correct statistical analysis used to support conclusions?

-Are there concerns about ethical or regulatory requirements being met?

Reviewer #1: Methodology has following limitations:

1. Snails were infected in a controlled lab environment, which might not fully represent natural infection dynamics in the wild.

2. Study not addressed how infection stage or parasite load affects the protein profiles detected by MALDI-TOF MS.

3. There is currently no established MALDI-TOF MS reference spectra database for *Schistosoma* parasites or infected snails, which is necessary for broader diagnostic use.

Reviewer #2: The methods are very clear.

Reviewer #3: (No Response)

Reviewer #4: Yes, The objectives of the study was clear to discriminate between the two species of Biomphalaria using MALDI-TOF MS and to assess whether MALDI-TOF MS could detect the S. mansoni infections in the snails.

Yes, it was well designed to address the stated objectives.

Yes, the field and experimental population were appropriate to test the hypothesis

Yes, the sample size is still acceptable but need to be increased in case of mass and field studies.

Yes, the statistical analysis was support the result and conclusions.

Yes, the protocol complies with national and international guidelines on animal welfare.

**Results:**

-Does the analysis presented match the analysis plan?

-Are the results clearly and completely presented?

-Are the figures (Tables, Images) of sufficient quality for clarity?

Reviewer #1: Results are properly presented

Reviewer #2: Result is aligning with the material and methods.

Reviewer #3: (No Response)

Reviewer #4: Yes, the results were well-presented and matched the study plan. The figures were of sufficient quality and clarity. Could the author able to add MALDI-TOF MS profiling with spectra and dendrogram for the uninfected and infected snail with S. mansoni.

**Conclusions:**

-Are the conclusions supported by the data presented?

-Are the limitations of analysis clearly described?

-Do the authors discuss how these data can be helpful to advance our understanding of the topic under study?

-Is public health relevance addressed?

Reviewer #1: Following limitations are noticed in the present study:

1. The current MALDI-TOF spectral database is in-house and may lack comprehensive reference spectra from diverse Schistosoma strains or Biomphalaria species, potentially limiting diagnostic accuracy.

2. MALDI-TOF MS might have limitations in detecting very early-stage or low-intensity Schistosoma infections in snails, which could affect sensitivity in real-world surveillance.

Reviewer #2: Conclusion is also very clear.

Reviewer #3: (No Response)

Reviewer #4: Yes, the conclusion was supported the data presented and the limitations were described in the abstract, method and conclusion. The authors discuss how his findings will help the field diagnosis of schistosomiasis. The public health concern was also addressed.

**Editorial and Data Presentation Modifications?**

Reviewer #1: (No Response)

Reviewer #2: No

Reviewer #3: (No Response)

Reviewer #4: Accept with minor revision.

**Summary and General Comments**

Reviewer #1: (No Response)

Reviewer #2: (No Response)

Reviewer #3: (No Response)

Reviewer #4: The manuscript presented robust MALDI-TOF MS data for the field and laboratory Biomphalaria species with clear objective of use these analysis for detection of S. mansoni infections in the snail to help in early diagnosis of schistosomiasis. The study is novel with significant impact in both animal and public health concern.

Few comments needed to be addressedby the authors;

1. The author will require to add MALDI-TOF MS profiling with spectra and dendrogram for the uninfected and infected snail with S. mansoni.

2. Fiqure (4) require to be edited accordingly. The alphabetic annotations need to be in brackets to have a space between the alphabetic and the B of Biomphalaria to be more clear for reader. Correct also the alphabetics to be A, B, C and D.

3. The discussion need to have more references from literature supporting the finding even from different parasite or different snail species.

PLOS authors have the option to publish the peer review history of their article (what does this mean? ). If published, this will include your full peer review and any attached files.). If published, this will include your full peer review and any attached files.

**Do you want your identity to be public for this peer review?** For information about this choice, including consent withdrawal, please see our For information about this choice, including consent withdrawal, please see our Privacy Policy ..

Reviewer #1: No

Reviewer #2: No

Reviewer #3: No

Reviewer #4: No

**Figure resubmission:** While revising your submission, we strongly recommend that you use PLOS’s NAAS tool (https://ngplosjournals.pagemajik.ai/artanalysis) to test your figure files. NAAS can convert your figure files to the TIFF file type and meet basic requirements (such as print size, resolution), or provide you with a report on issues that do not meet our requirements and that NAAS cannot fix.

**Reproducibility:** To enhance the reproducibility of your results, we recommend that authors of applicable studies deposit laboratory protocols in protocols.io, where a protocol can be assigned its own identifier (DOI) such that it can be cited independently in the future. Additionally, PLOS ONE offers an option to publish peer-reviewed clinical study protocols. Read more information on sharing protocols at https://plos.org/protocols?utm_medium=editorial-email&utm_source=authorletters&utm_campaign=protocols To enhance the reproducibility of your results, we recommend that authors of applicable studies deposit laboratory protocols in protocols.io, where a protocol can be assigned its own identifier (DOI) such that it can be cited independently in the future. Additionally, PLOS ONE offers an option to publish peer-reviewed clinical study protocols. Read more information on sharing protocols at https://plos.org/protocols?utm_medium=editorial-email&utm_source=authorletters&utm_campaign=protocols

---

## [Decision Letter · Decision Letter 1]

14 Dec 2025

Thank you for submitting your manuscript to PLOS Neglected Tropical Diseases. After careful consideration, we feel that it has merit but does not fully meet PLOS Neglected Tropical Diseases's publication criteria as it currently stands. Therefore, we invite you to submit a revised version of the manuscript that addresses the points raised during the review process.

* A rebuttal letter that responds to each point raised by the editor and reviewer(s). You should upload this letter as a separate file labeled 'Response to Reviewers '. This file does not need to include responses to any formatting updates and technical items listed in the 'Journal Requirements' section below.'. This file does not need to include responses to any formatting updates and technical items listed in the 'Journal Requirements' section below.

* A marked-up copy of your manuscript that highlights changes made to the original version. You should upload this as a separate file labeled 'Revised Manuscript with Track Changes '.'.

* An unmarked version of your revised paper without tracked changes. You should upload this as a separate file labeled 'Manuscript '.'.

We look forward to receiving your revised manuscript.

Kind regards,

Academic Editor

Paul Mireji

Section Editor

Shaden Kamhawi

co-Editor-in-Chief

Paul Brindley

co-Editor-in-Chief

**Journal Requirements:**

At this stage, the following Authors/Authors require contributions: Doudou Sow. Please ensure that the full contributions of each author are acknowledged in the "Add/Edit/Remove Authors" section of our submission form.

2) Please amend your detailed Financial Disclosure statement. This is published with the article. It must therefore be completed in full sentences and contain the exact wording you wish to be published.

**Reviewers' comments:**

Reviewer's Responses to Questions

**Key Review Criteria Required for Acceptance?**

**Methods**

-Are the objectives of the study clearly articulated with a clear testable hypothesis stated?

-Is the study design appropriate to address the stated objectives?

-Is the population clearly described and appropriate for the hypothesis being tested?

-Is the sample size sufficient to ensure adequate power to address the hypothesis being tested?

-Were correct statistical analysis used to support conclusions?

-Are there concerns about ethical or regulatory requirements being met?

Reviewer #2: -Are the objectives of the study clearly articulated with a clear testable hypothesis stated?

yes

-Is the study design appropriate to address the stated objectives?

yes

-Is the population clearly described and appropriate for the hypothesis being tested?

yes

-Is the sample size sufficient to ensure adequate power to address the hypothesis being tested?

yes

-Were correct statistical analysis used to support conclusions?

yes

-Are there concerns about ethical or regulatory requirements being met?

yes

Reviewer #3: No. The author has not responded to the concerns raised by me nor have addressed them in the revised manuscript.

Reviewer #4: (No Response)

**Results**

-Does the analysis presented match the analysis plan?

-Are the results clearly and completely presented?

-Are the figures (Tables, Images) of sufficient quality for clarity?

Reviewer #2: yes

Reviewer #3: No

Reviewer #4: (No Response)

**Conclusions**

-Are the conclusions supported by the data presented?

-Are the limitations of analysis clearly described?

-Do the authors discuss how these data can be helpful to advance our understanding of the topic under study?

-Is public health relevance addressed?

Reviewer #2: yes

Reviewer #3: Somewhat

Reviewer #4: (No Response)

**Editorial and Data Presentation Modifications?**

Reviewer #2: no

Reviewer #3: Major revision

Reviewer #4: (No Response)

**Summary and General Comments**

Reviewer #2: NA

Reviewer #3: The authors have not addressed the concerns raised by me

Reviewer #4: (No Response)

PLOS authors have the option to publish the peer review history of their article (what does this mean? ). If published, this will include your full peer review and any attached files.). If published, this will include your full peer review and any attached files.

**Do you want your identity to be public for this peer review?** For information about this choice, including consent withdrawal, please see our For information about this choice, including consent withdrawal, please see our Privacy Policy ..

Reviewer #2: No

Reviewer #3: No

Reviewer #4: **Yes:** Mohamed A. HelalMohamed A. Helal

**Figure resubmission:**
---

## [Editor Report · Decision Letter 2]

3 Mar 2026

Dear Pr Ranque,

We are pleased to inform you that your manuscript 'Rapid identification of Biomphalaria spp. and diagnosis of Schistosoma mansoni infestation using MALDI-TOF mass spectrometry' has been provisionally accepted for publication in PLOS Neglected Tropical Diseases.

Best regards,

Anand Setty Balakrishnan, PhD

Academic Editor

Paul Mireji

Section Editor

Shaden Kamhawi

co-Editor-in-Chief

Paul Brindley

co-Editor-in-Chief

---

## [Editor Report · Acceptance letter]

Dear Pr Ranque,

We are delighted to inform you that your manuscript, "Rapid identification of Biomphalaria spp. and diagnosis of Schistosoma mansoni infestation using MALDI-TOF mass spectrometry," has been formally accepted for publication in PLOS Neglected Tropical Diseases.

Best regards,

Shaden Kamhawi

co-Editor-in-Chief

Paul Brindley

co-Editor-in-Chief
